# Contextualizing Corruption: A Cross-Disciplinary Approach to Studying Corruption in Organizations

**Kanti Pertiwi** [1,2]

1   Department of Management, Universitas Indonesia, Jawa Barat 16424, Indonesia; kanti.pertiwi@ui.ac.id or kanti.pertiwi@unimelb.edu.au
2   Department of Management & Marketing, The University of Melbourne, Parkville VIC 3010, Australia

**Abstract:** This paper aims to establish how organization and management research, an extensive field that has contributed a great deal to research on corruption, could apply insights from other disciplines in order to advance the understanding of corruption, often considered as a form of unethical behavior in organizations. It offers an analysis of important contributions of corruption research, taking a 'rationalist perspective', and highlights the central tensions and debates within this vast body of literatures. It then shows how these debates can be addressed by applying insights from corruption studies, adopting anthropological lens. The paper thus proposes a cross-disciplinary approach, which focuses on studying corruption by looking at what it means to individuals implicated by the phenomenon while engaging in social relations and situated in different contexts. It also offers an alternative approach to the study of corruption amidst claims that anti-corruption efforts have failed to achieve desirable results.

**Keywords:** organizational corruption; business ethics; management; governance

## 1. Introduction

'Corruption' has largely been construed as an undesirable and destructive aspect of social life. There are deeply rooted notions about 'corruption' as 'decay' or 'impurity' (Hindess 2012). Consequently, throughout modern Western history, corruption has been deemed to be the enemy of humanity. Many social institutions such as governments, educational and religious foundations, as well as the media, articulately condemn corruption as malignant and align their policies with such a disposition. These policies often include various anti-corruption measures as well as good governance principles, codes and the alike, which are all produced with the aim of abolishing corruption. Yet 'corruption' has made its entrance into the lives of people in different societies and cultures.

'Anti-corruption' arguably entered the scene of international development in the late 1990s in what Naim (1997) called the 'corruption eruption'. There was an overwhelming call, locally and globally at that time, for the eradication of corruption. This call was led by international development agencies, particularly the World Bank (Koechlin 2013). This growing emphasis on the negative effects of corruption led to significant efforts within the research community to unpack the complexities of corruption and identify the ways through which it might be completely eradicated from human interactions. A quick research on the *Web of Science* portal reveals that there was a significant increase in the number of studies on corruption, starting with 1125 articles in the year 2000 but increasing to 18,604 academic articles published by end of year 2017. Scholars in the field of management and organization, like other social scientists, took great interest in examining corruption. Their efforts were prompted by the surge of various scandals involving various business or government organizations around the globe.

Management and organization studies mostly considered corruption as organization misbehaviour (Ackroyd and Thompson 1999), a type of crime (Aguilera and Vadera 2008), the dark side of organizations (Linstead et al. 2014). They also viewed corruption in rationalistic terms in that they perceived corruption as the result of rational agents exercising their rational thinking so as to maximize individual gains. However, this perspective detached the individual from his or her social relations and circumstances. Moreover, it also viewed corruption as an 'objective' fact of life and sought to uncover its true causes and consequences (Sonenshein 2007; Martin and Parmar 2012). A deeper examination of the works of management and organization scholars reveals that there are still debates in the literature pertaining to corruption which need to be addressed. These debates concern whether corrupt behavior should be considered mindful or 'mindless', the extent to which social dimensions influence individuals engaging in corruption and whether ethical issues associated with corruption are 'given' and objectively identifiable or are constructed by individuals in specific social contexts. My analysis of these debates suggests the need for research to look at how corruption is interpreted by actors engaging in social relations and situated in a particular context. It is useful to view these debates by applying insights from anthropological studies on corruption because this approach highlights the need to study corruption using a cross-disciplinary approach outlined in this paper.

This paper is organized as follows: First, it will review some important works on corruption which I label as 'rationalist approaches'; approaches which are often adopted in various fields, including in the area of management and organization. This covers a vast body of literature that has contributed significantly to our understanding of corruption from organization and behavioral perspectives. Second, the paper will then present the three central debates within this particular literature. Third, in addressing these debates, the paper will draw insights from anthropological and related studies to suggest a cross-disciplinary approach to researching corruption. To conclude, it will highlight potential contributions of such an approach.

## 2. Review of Literature

### 2.1. Unpacking Corruption: Rationalist Approaches

One dominant approach to studying corruption might be termed 'rationalist'. This includes theory and research that takes both a macro and a micro perspective. The macro (i.e., country-level) view has been adopted by many scholars in law, economics and politics, looking at corruption and its effects on a host of variables such as a country's political processes, economic performance and other measures of development. The micro perspective has been adopted, in particular, by management and organization scholars who discuss corruption as a type of unethical behavior which may be analyzed at individual and organization levels. Both perspectives tend to assume that corruption is in and of itself inherently harmful to society. They also regard it as behaviourally dysfunctional. Central to these rationalist views is the assumption that corrupt individuals are rational actors seeking to maximize their gains. I will describe each of these perspectives and the findings that have been generated from these assumptions.

'Rationalist' research maintains that corruption is in and of itself inherently harmful or dysfunctional to society and many scholars describe it in negative terms (Torsello and Venard 2016) as a generic 'social problem'. They commonly argue that corruption hurts economic growth and retards development. Adopting the World Bank's definition of corruption as the 'abuse of public office power for private gain', they adopt the public-private dichotomy that underpins much of the mainstream corruption research. These scholars assume that there is a similar division in markedly different societies with contrasting cultures between what is considered as public and private goods. Meanwhile other studies have shown that these important factors, rather than being universal, are historically determined and locally specific (Rothstein and Torsello 2014).

The rationalist perspective further maintains that corruption is detrimental to investment, productivity (Lambsdorff 2003) and, therefore, a country's economic growth rate (Mauro 1995). It

has been argued that its effects are weaker in the less developed nations, possibly because the scale and type of corruption found there is considered 'more predictable' when corrupt governments behave as expected by those seeking favors. Hence there is less negative impact on investment (Campos et al. 1999). Other rationalists contend that corruption leads to the unfair allocation of resources and a poor quality of infrastructure (Klitgaard 1988). At the same time, they speculate that this hinders a firm's growth because paying bribes increases costs but does not always guarantee an increase in profits (Fisman and Svensson 2007).

Other research has found that corruption is inversely linked to the degree of democracy. Countries which have fully democratized have lower levels of corruption than those only partially democratized because of the lack of competition between political actors (Montinola and Jackman 2002). These authors contend that in fully democratized countries, officials or politicians have lower incentives to engage in bribe-taking because they can be replaced rather easily by their constituents through democratic processes. Countries which are considered more democratic have lighter regulation for entry for start-up firms thus lower levels of corruption (Djankov et al. 2002) due to the assumption that more democratic governments face more pressures to not create burdensome regulations. Finally, when looking at the quality of democratic institutions, which is the extent to which there is competition and openness in the electoral systems, Bhattacharyya and Hodler (2010) maintain that corruption is higher in cases where the quality of democratically controlled institutions is below a certain threshold. They argue that it is inversely lower where these institutions are stronger because they are effective barriers to a government's and politicians' rent-seeking activities.

There are some counter-arguments to this negative view of corruption. For example, Lui (1985) proposes that bribery 'greased the wheels' of the economy, therefore benefitting governments. Meon and Weill (2010) also argue that corruption is beneficial in a weakly governed country, particularly where governments are considered ineffective and prone to producing burdensome regulations. Corruption, this argument runs, helps economic growth in these countries but can prove costly in others which do not suffer weak governance. Similarly, a recent study by Huang (2016) which looks at 13 countries in the Asia Pacific using data from 1997–2013 challenges the conventional wisdom that corruption is bad for economic growth. The author contends that corruption plays a positive role in stimulating growth in South Korea while it has had an adverse effect on growth levels in China, suggesting there is not a universally linear relationship between the two variables.

Some researchers stress that corruption can be seen as either 'dysfunctional' or 'functional', depending on the institutional settings. This points to the importance of considering the corresponding political and economic systems as well as the cultural and legal environments (Girling 1997; dela Rama and Rowley 2017). A related body of literature discusses 'state capture'—how businesses capture the state by making private payments in order to influence laws, rules, decrees or regulations. 'State capture'—or corruption—is beneficial for the captor firms' performance but detrimental for the rest of the economy (Hellman et al. 2003; Rijkers et al. 2017). Recent work supports this view by questioning the extent to which corruption harms as opposed to benefits a firm's competitive position. Instead of viewing corruption as inherently destructive, the corporate political strategy literature suggests that corruption benefits corporations by way of developing political ties and exploiting regulatory processes (Galang 2012; Nguyen et al. 2016). For example, some studies have looked at how former politicians or cabinet members are recruited as board members, suggesting that firms are increasingly aware of the benefits of having political ties to influence policy and regulations (Hillman 2005; Lester et al. 2008; Zheng et al. 2015).

As mentioned, rationalist scholars adopting a macro view also believe that corruptors are rational actors in that corruption results from a rationally calculated cost and benefit analysis on the part of the party committing it. As long as the benefit of corruption exceeds the costs, corruption continues. Thus some scholars argue that business-government corruption can be eliminated by increasing competition between firms within markets as this will increase the cost of paying bribes (Ades and Di Tella 1999) although evidence from post-communist countries suggests otherwise (Diaby and Sylwester 2015).

In a similar vein, others suggest that government wages must be increased—so that bribe-payers would have to increase their offerings if they are to compete with legitimate earnings (Van Rijckeghem and Weder 2001; An and Kweon 2017).

Overall, despite their contribution, the works cited above have received much criticism. For instance, the rational economic view of corruption has been deemed 'too narrow and too narrowly technical' (Hindess 2012). Moreover, these studies assume that corruption is universally harmful or dysfunctional (Harrison 2006). They also assume that corrupt individuals are rational actors. Therefore, to control corruption, conditions must be created in which the costs of engaging in corruption exceed the benefits. As a result, these views tend to ignore the complexities of norms and cognitions (Misangyi et al. 2008), which is the focus of management and organization scholars whose work I discuss next.

## 2.2. Rationalist Works in Organization and Management Studies on Corruption

The management and organization literature discusses corruption or unethical behavior both at individual and organization levels. Corruption has been studied as a particular form of unethical behavior, which harms the organization and the society as a whole (Cleveland et al. 2009; Rose-Ackerman and Palifka 2016). Many of these studies are built upon the assumption that corruption occurs due to some kind of moral deficiency located within self-interested individuals (Bracking 2007; Gyekye 2015).

Researchers interested in unpacking corrupt behavior employ a variety of methods, including experiments, interviews of different kinds, and narrative analysis. In so doing, various explanations have emerged either focusing on the idea that corruption arises because of 'bad apples' such as corrupt individuals, or because of 'bad barrels' as in certain types of organizations which encourage corruption. Extending the 'bad barrels' argument, scholars highlighted the importance of understanding the 'bad larder' (Gonin et al. 2012) or the context of the organization and its influence on corruption. I will begin by summarizing the findings from this body of literature under the metaphors of 'bad apples', 'bad barrels' and 'bad larders'. I then identify three emerging debates emanating from these discussions. Finally, I conclude that it is necessary to view corruption through a different lens to properly address the issues raised through these debates.

The 'bad apples' argument stresses that unethical behaviors in organizations are due to the personal characteristics of differing individuals (Brass et al. 1998). In other words, some people are just born 'bad' or raised to be 'bad' and they are unable to stop themselves doing bad things (Fleming and Zyglidopoulos 2009). For example, individuals are more likely to engage in corrupt behavior when they are ambitious (Jackall 1988) or have a stronger external locus of control—the tendency to assign responsibility for a situation to something beyond the control of the individual (e.g., Reiss and Mitra 1998). Others maintain that those who have a relativistic morality (that is situation-dependent) as opposed to idealistic (universal morality) (e.g., Elias 2002); or have low empathy with others' situation (Detert et al. 2008) are more prone to corruption than those who do not. Other findings suggest that better ethical decisions are made by females compared to males, by older people compared to younger people (O'Fallon and Butterfield 2005), and by people who are more religiously committed compared to those who are not (Singhapakdi et al. 2000). Initially, it was also argued that women appear to be less tolerant of corruption than men, especially in Western culture (Alatas et al. 2009) while a more recent study found that women's representation in government reduces corruption (Esarey and Schwindt-Bayer 2017). More recently, using the organization identification perspective, Vadera and Pratt (2013) argue that individuals who over-identify—have a sense of strong attachment to the organization—are more likely to commit corrupt acts with the intention of benefitting the organization. Others observe that people from a certain cultural milieu, such as India, are more tolerant of corruption than others, such as people from Australia, while in the case of Singapore and Indonesia, people are found to be more and less tolerant than expected, respectively

(Cameron et al. 2009). Still others suggest that lower levels of perception of corruption are found in more individualistic compared to collectivist countries (Jha and Panda 2017).

While the 'bad apples' argument draws attention to the role of individual attributes, the 'bad barrels' argument highlights features of the organization in facilitating corruption. These arguments complement and, at the same time, challenge the previous 'bad apples' argument. In the first instance, they question the ability of individuals to escape from corruption as well as the role of cognition and ethical reasoning in deciding the agent's responses. Second, they acknowledge the possibility that even 'good apples' might engage in corruption and develop 'mental strategies' to cope with the possible dissonance felt after committing a questionable act (Fleming and Zyglidopoulos 2009). This might, for example, involve producing an account which helps one to feel better about acting corruptly. Instead of viewing corruptors as individuals having perfect agency, the proponents of the 'bad barrel' argument suggest that corruption occurs due to factors within the organization, including the organization's ethical climate, culture and leadership.

Ethical climate is the collective organizational normative structure (Victor and Cullen 1988) which influences ethical decision making. An egoistic climate, for example, correlates positively with unethical behavior (Peterson 2002), and more specifically corruption (Gorsira et al. 2018), while a positive ethical climate has a positive influence on ethical behavior (O'Fallon and Butterfield 2005) through collective empathy—that is caring about others likely to be affected by the behavior, and a sense of a collective efficacy—the belief that the behavior will have the desired effect (Arnaud and Schminke 2012). An ethical culture can also reduce unethical behavior (Schaubroeck et al. 2012). Culture refers to formal (e.g., reward systems, ethics training programs) and informal systems such as peer behavior and identity-building stories (Schaubroeck et al. 2012).

Through practising ethical leadership, a set of traits that will promote the development of a shared understanding of what constitutes an ethical culture, unethical behavior such as corruption can be reduced. This is consistent with the findings that when an organization's leaders are perceived to be ethically positive, there are lower reports of counterproductive employee behavior (Mayer et al. 2009). One of the ways to promote the shared understanding is to tell powerful stories about ethics which others can replicate, or through delivering formal speeches in order to communicate organization's expectations (Schaubroeck et al. 2012). In contrast, when leaders downplay the negative consequences of misconducts, or in other words they become morally disengaged, employees' ethical behaviour is negatively affected (Bonner et al. 2016).

The extension of the 'barrel' allegory is the 'bad larder' (Gonin et al. 2012), which refers to factors outside the organization, such as the industry culture or climate, network relationships, the role of government and societal norms or values. This argument stresses that corruption often occurs due to certain inter-firm practices such as gift-giving (Verhezen 2009), or networking activities between business and government that can potentially turn into corruption (La Porta et al. 1999). Densely connected subgroups—referred to as cliques (Doreian 1971)—are able to develop and sustain distinct subgroup cultures and norms which support corruption (Brass et al. 1998). Furthermore, cliques operate under advance mechanisms in which a dense network of relationships between individuals and organizations facilitate illegal activities covered by legal ones such as using one's expertise and professional knowledge to mask illegal deals and decisions (Jancsics and Javor 2012). Similarly, in the field of political sociology, it was argued that the presence and persistence of informal ties referred to as 'cliques' are associated with potential misconduct or procedural irregularities (Ozierański and King 2016).

Focusing more on relationships, scholars argue that relationships lead to corruption when there is a felt obligation to reciprocate others' treatment (Palmer 2008). Moreover, language becomes an important facilitator in helping individuals understand interactions in reciprocal relations; naming a gift as a 'bribe' signals higher expectation for reciprocity (Lambsdorff and Frank 2010). Other scholars have studied the role of government whereby more intrusive regulations (Treisman 2007) and more ties to government agents increase the likelihood of firms opting to bribe because these ties assist

managers in undermining rules regarding questionable practices (Collins et al. 2009). Looking at the influence of social norms on corruption, two norms are particularly relevant: reciprocity and a high achievement orientation. The former makes firms' managers more tolerant to exchanging favors which may have ethical implications (McCarthy et al. 2012), while the latter makes an organization become more prone to bribery (Martin et al. 2007).

Integrating 'bad apples, bad barrels, and bad larders', some scholars argue that corruption or unethical behavior is a result of an individual's deliberation which, in turn, is an outcome of his or her responses to situational factors (Trevino 1986). This explains why moral cognition does not always end in moral action, as certain situations may influence an individual's final decision. Drawing from Kohlberg (1969) and others, Trevino proposed the 'person-situation' model in which an individual's evaluation of right or wrong is moderated by individual moderators such as the strength of their ego, independence in the field and locus of control, as well as the situational moderators arising out of their cultural and job-related context.

Ego strength refers to how strongly a person follows his or her convictions and rejects impulses. Field dependence refers to the degree of reliance on external referents to guide decision-making, and locus of control refers to the general belief of individuals about whether they have control over life events or whether things happen beyond their control (Trevino 1986). Situational moderators include whether the organization has a clear position about right and wrong and which behavior will be rewarded and which will be punished (O'Fallon and Butterfield 2005; Lehnert et al. 2015). In addition, other external pressures such as the pressure to make decisions concerning competitive positions under time constraints, also influence behavior.

Similarly, Jones (1991) argues for an issue-contingent model which regards unethical behavior as issue-dependent. Like Trevino, Jones' model contends that decision making is partly determined by social learning within the organization (Loe et al. 2000; MacDougall et al. 2015). An individual's engagement in (un)ethical behavior is partly influenced by the intensity of the issue in that an issue which is morally more intense will lead to more ethical decisions. Hunt and Vitell's (1986) theory of marketing ethics offers a similar perspective by including not only individual variables but also the environment which consists of organizational, industry, and cultural norms. They argue that norms determined by social consensus or demonstrated by leaders influence individuals' ethical judgment.

The idea that decent people can engage in corruption if they are caught up in a difficult situation or environment can be explained by the concept of rationalization—the 'mental strategy' that individuals develop to cope with any dissonance they might experience in engaging in corruption, which in turn assists in making corruption seem 'normal'— in other words, normalizes corruption (Ashforth and Anand 2003; Lennerfors 2017).

The rationalist literature speaks of corrupt individuals as having a psychological mechanism that allows them to neutralize any negative feelings that result from engaging in corrupt acts. It involves the effort to construct a narrative which justifies an act that would originally be questionable (Fleming and Zyglidopoulos 2009). In his analysis of the accounts of Abramoff, an American lobbyist charged for a wide range of corrupt actions, Gray (2013) discusses several techniques that are used around lobbying activities, namely indirect gifting—giving congressmen money through "fundraisers", revolving doors—which involves the circulation of congressmen to lobbying posts, and devising a situation that supports rationalization on the part of the officials. Rationalization or neutralization strategies (Gray 2013) may seem to emphasize the idea of agency. However, authors in this stream assert that individuals rationalize not in isolation, but in relation to their social settings. Scholars have identified several rationalization strategies (Ashforth and Anand 2003), which includes softening the immorality of their act by using euphemisms or metaphors such as "fighting in a war" to justify questionable actions (Campbell and Göritz 2014). Empirical research supports the idea that euphemisms are used to make corruption more acceptable (Znoj 2007) by putting the blame on others (a strategy called 'denial of responsibility'). This can involve, for example, actors who make corrupt

payments labelling these as extortion which had to be paid. Actors may also deny causing any injury by engaging in narratives such as "no one is affected" or "it's a small payment, just for expediency".

Ethical distance (Zyglidopoulos and Fleming 2008)—which refers to the distance between one's act and its consequences—is useful in explaining systemic corruption—the kind of corruption that is said to be common in non-Western societies (Breit and Vaara 2014). Researchers argue for two types of distance: Temporal and structural. In each type, an accompanying rationalization may be activated. In temporal distance, individuals perceive that corrupt acts have no immediate effect because no penalty has ever beset the individual or the organization using in it, therefore engaging in corruption is not so perplexing. The rationalization that may be triggered in this case is, for example, the denial of injury—"it does not hurt anybody". In structural distance, individuals are insulated from the sense of moral obligation of corruption because they see their role in it as a small part of a larger whole. Within the organization, the individuals perceive that moral obligation is distributed amongst the individuals involved, which means the more people involved the easier it is to escape any moral burden. In collective systemic corruption, individuals perceive their practice as no different to others' so it reduces the dissonance that may surface. In this case, the rationalization that is being triggered is, for example, "everybody's doing it".

## 3. Emerging Debates in Management and Organizational Corruption Research

This review has so far shown how corruption is understood using different concepts and approaches within the 'rationalist' literature. I will now focus on three key debates emanating from the above discussion. The first debate considers whether ethical behavior (or unethical behavior such as corruption) is mindful or mindless, the second examines whether unethical decision makers are discrete individuals or embedded in a social context, and the third explores whether ethical issues such as corruption are objective or constructed. Each will be discussed in turn, starting with an explanation of the debate, followed by relevant theories and empirical support, and concluding with a discussion of how these debates point to the value of bringing in anthropological approaches in studying corruption.

### 3.1. (Un)ethical Behavior: Mindless or Mindful

The first debate questions the assumptions of the rationalistic approach to corruption and considers whether corruption should be assumed to be a mindful act or whether scholars should consider the possibility that corrupt behavior flows from mindlessness. Mindfulness or heedfulness (Weick and Roberts 1993) refers to the state of being careful, critical, purposeful, attentive and vigilant, akin to the condition required in being rational or using reason: The individual has intent, is putting in effort, and able control the process (Bargh 1994). Mindlessness is characterized as non-conscious processing of repetitive behavior (Ashforth and Fried 1988; Smith-Crowe and Warren 2014), representing "a failure to see, to taken note of, to be attentive to" (Weick and Roberts 1993, p. 61) what is going on. Similarly, intuition is used in describing the psychological process that occurs "quickly, effortlessly, and automatically, such that the outcome but not the process is accessible to consciousness" (Haidt 2001, p. 818).

When individuals act mindlessly, they act "with little or no real problem solving or even conscious awareness" (Ashforth and Anand 2003, p. 14), therefore the corrupt act is not an outcome of moral reasoning, a process which is intentional and effortful (Langer and Moldoveanu 2000). Mindlessness can occur due to social influence and organizational structures (Palmer 2008). Social influence includes the authorization of corruption by leaders, the socialization of corruption itself or an escalation of commitment, in which organization members engage in corruption to reduce dissonance over past decisions which subsequently appear to lack merit (Palmer 2008; Staw 1976). For example, instead of trying to rectify a decision that is later found to be defective, organization members increase their commitment towards the decision in question, simply because they want to avoid continued dissonance (Palmer 2008).

Social influence processes such as general consensus puts pressure on individuals to believe that their decisions are meritorious, while organizational structures limit individual capacity to make the right call concerning ethical issues. Organizational structures refer to how tasks are distributed across different parts of the organization as well as the routines developed to guide these tasks. For example, the recall division at Pinto (the car company which failed to recall faulty products in the 1990s) was separated from its safety test division in such a way that the company's information flow was badly managed, which subsequently impaired decision making. In other words, corruption is enacted mindlessly because people experience pressures from their superiors or peers, or because people are 'locked' in within certain organizational rules, scripts and schemas which make them 'fail' to deliberate and choose a different course of action (Palmer 2008).

Rather than seeing corrupt acts as the outcome of deliberate 'mindful' reasoning, some scholars argue it is more likely to be the result of mindlessness (Sonenshein 2007). Social psychological research notes that "moral reasoning is rarely the direct cause of ethical judgment" (Haidt 2001, p. 815). Individuals' ethical or moral judgment is instead derived from a quick evaluation or intuition, which in turn is influenced by social and cultural factors (Haidt 2001). Scholars question whether rationalization precedes corrupt behavior, as opposed to occurring after the act and there appears to be no relationship between rationalization strategies and the desire or the intention to act corruptly (Rabl and Kuhlmann 2009). If mindlessness really prevails and rationalizations only occur after the fact, implications exist for the way scholars study corruption. Furthermore, Palmer's (2008) thick descriptions of corruption narratives and detailed information of actors' thought and emotions, show that there may be alternative explanations of corruption as a result of mindless as opposed to mindful processing.

### 3.2. Ethical Behavior: Atomistic or Embedded

The second debate promotes the idea of exploring the notion of the 'barrel' or 'larder' more deeply. It highlights that, instead of treating corruption in isolation from its context, scholars should give more attention to social aspects of corruption as well as to how social relations influence the meanings of corrupt practices (Misangyi et al. 2008). Business ethics researchers in particular tend to overlook the effect of social factors in ethical decision making (Bartlett 2003). Therefore, researchers argue that factors such as business culture, industry characteristics or societal norms demand greater consideration. For instance, unethical practice is influenced by a weak business culture which tends to lead to non-transparent practices and strong potentially corrupt connections between business and politicians) (Vaiman et al. 2011). A market that is characterized by concentrated ownership of firms in the hands of a number of wealthy families similarly encourages rent-seeking behaviors between businessmen and the government (Fogel 2006). Others suggest that high scores in the cultural dimension of power distance (the extent to which people accept an unequal distribution of power) and masculinity (the extent to which people stress materialism and wealth) correlate with corruption (Getz and Volkema 2001).

The above assertions seem to have only scratched the surface of what other scholars refer to as social context. These other scholars suggest that explanations for corruption lie beyond culture or structure and that they are intrinsically bound up with the meanings and identities of people and their practices (Misangyi et al. 2008). These meanings and identities are reproduced in ongoing social relations (Sewell 1992), shaped by interactions between social actors who continuously interpret, carry out and enact them (Zilber 2002). They are also the "way(s) of how a particular social world work" (Jackall 1988, p. 112). In other words, the meanings and identities are the 'driving forces' for behavior and they have rarely been explored by corruption researchers.

Seeing corruption as embedded in meanings and identities is particularly important in the case of systemic or institutionalized corruption (Misangyi et al. 2008), where corruption is widespread and treated as legitimate or no longer questioned. Misangyi and colleagues (Misangyi et al. 2008) argue that in systemic corruption, corrupt practices are interpreted differently by individuals. Therefore, to

change an already corrupt system one needs to change the meanings assigned to the practices within that system.

*3.3. Ethical Issues: Objective or Constructed*

The third debate in the literature questions the claims of rationalist researchers that corruption is objectively identifiable and takes the idea of meaning even further to suggest that (un)ethical or deviant behavior (such as corruption) is socially constructed. Scholars have acknowledged the importance of decision makers' perceptions in deciding to engage in particular actions. For instance, individuals' perception of uncertainty within their environment will have an impact on internal and external networking activities (Sawyerr 1993) which may include ethically questionable practices such as gratuity and bribery (Mele 2009). Similarly, managers' perceptions of financial constraints and of competition intensity in a market influence firms' decision to bribe (Martin et al. 2007). This shows that it is important to account for how firms interpret or perceive their environment.

Aside from arguing that interpretation of decision-making variables varies, some scholars have also acknowledged the importance of actors' perceptions in determining whether the behavior under study constitutes 'misbehavior', 'deviance' or indeed 'corruption'. Scholars who argue for this view make largely objectivist assumptions—that individuals interpret their environment in a similar manner and that they are uncovering cues from their environment as opposed to actively constructing their own situations or problems. Martin and Parmar (2012) further contend that interpretation works in a more complex way than what is described in rationalist studies. Rationalist corruption studies rarely problematize the possibility of a more varied interpretation of the proxies for 'cultural practices', 'financial constraints', 'competition' and 'government intervention' in their survey items.

On the other hand, few corruption studies are convinced that individuals are not passive but active interpretive actors, acknowledging the varied interpretations of human problems and conditions by individuals (Weick 1979; Berger and Luckman [1967] 1971). Sonenshein (2007), for example, questions the rationalist models described above and contends that individuals construct ethical issues in a much more nuanced way, producing "more idiosyncratic interpretations" (Sonenshein 2007, p. 1029), often with very limited information, and make ethical judgments intuitively as opposed to rationally, with less deliberation than scholars have generally believed. Consistent with Haidt (2001), he argues that moral reasoning is used only after decisions are made, partly to help individuals justify the decisions or to explain for the rapid processing beyond their awareness that occurs prior to facing the decisions' outcomes.

In constructing issues, people draw on: (1) Social anchors (communicating with other individuals) to interpret the moral intensity of an issue, and (2) their understanding of others' interpretation of an issue by forming a mental model. These two mechanisms highlight that issue construction is not only individual but also social. Moreover, issues are to be understood in a much more nuanced way, as opposed to being treated as binaries, i.e., 'triggering ethical dilemma' or 'not triggering ethical dilemma'. Individuals do not merely react to stimuli, they construct meanings (Boland and Tenkasi 1995).

This idea that individual construction or interpretation varies is supported by Turgeman-Goldschmidt (2008) who studied the life experiences of a group of computer hackers and illustrated how individuals assigned meanings to practices which did not correspond with the 'unethical' or 'deviant' label used in rationalist research. Commonly perceived as a specific type of computer-related deviance, hackers in their study actively constructed a positive identity for themselves by arguing that, for example, they were creating a 'better world' by 'not letting companies like Microsoft control the market', or perceiving themselves as a 'guardian of the state' by invading computer systems of the state's enemy. Similarly, Walton (2013b) has found that instead of seeing practices of *wantok*—an informal exchange between people from the same clan or family often

associated with nepotism as destructive, people in Papua New Guinea see them as "social protection mechanisms" (Walton 2013b, p. 187), because they help pull people out of poverty.[1]

These findings suggest that what outsiders label as 'unethical', 'deviant' or 'corrupt' may not be understood as such by the individuals concerned. This is why scholars have called on researchers to "study the interpretive processes" (Sonenshein 2007, p. 1026) through which individuals interpret or construct (un)ethical behavior such as corruption because of the multifarious and contested nature of the behavior.

Apart from the construction of issues surrounding corruption, the notion of 'ethics'—generally understood as individual's evaluation of good and bad—is also problematic because, similar to corruption, it has often been construed as objective as opposed to subjective and situated in a particular place and time. Recent scholarship argues that in order to understand ethics or morality, one needs to look at how issues pertaining to ethics or morality are constructed in social interactions of everyday life (Tileaga 2012).

In summary, some management and organization scholars have called for a more nuanced way of understanding the environment as part of the process of issue construction (Sonenshein 2007). Issue construction, further referred to as interpretation (Sonenshein 2007), is the process by which individuals create their own meaning by using stories or narratives as social events unfold (Boland and Tenkasi 1995). Because individuals construct an issue based on their expectation (what they expect to see) and motivations (what they want to see). Sonenshein (2007, p. 1026) suggested that researchers "study the interpretive processes that construct ethical issues out of social stimuli in the environment".

The last debate emanating from the literature in particular suggests that 'dysfunctional behavior' such as corruption has multiple meanings as it is socially constructed. Consequently, corruption needs to be studied in a way that can recognize and explore its social and varied construction. In this regard, I have drawn from anthropological research (Haller and Shore 2005; Torsello and Venard 2016) to study corruption which emphasizes its social, multifarious and contextualized meanings, an approach I now explain in more detail.

## 4. Anthropological Approaches to Corruption

In addition to the dominant rationalist approach to studying corruption, there is a growing and diverse body of research which looks at corruption based on a different set of assumptions. I use the term 'anthropological' approach to describe this work, although it is by no means a clear-cut body of literature and encompasses studies in fields covering not only anthropology but also sociology, human geography, discourse and human ethics.

The anthropology and sociology literature overlap in terms of their treatment of corruption as a social construction. However, further engagement with both literatures shows that they are often different in terms of the focus of their analysis and their theoretical orientation when analyzing corruption. For example, sociologists tend to be more interested in the 'causes and processes' (Hodgkinson 1997, p. 21), the structural elements (institutions, organizations and policy) or the macro-societal context and different scenarios of corruption (Numerato 2009), whereas anthropologists are less so. Instead, they tend to focus more on the meaning-making, also linguistic aspects of experiences of corruption, to which this paper draws attention, among others. As a result, there are more empirical materials from the anthropology literature that speak directly to the mainstream organizational literature, compared to the sociology literature. On the other hand, the field of anthropology itself is vast and can often be classified into two: Cultural and organizational anthropology, which are also different in regards to their level of analysis. Works in cultural

---

[1]　Of course readers may also argue that this meaning is mostly relevant to 'small' or 'petty' corruption involving everyday people as opposed to 'grand' corruption which implicates people in top positions in business and government. However, the extent to which certain meanings are only applicable for certain types of corruption has been debated by scholars, for example see Kennedy, D. 1999. The international anti-corruption campaign. *Connecticut Journal of International Law* 14: 455.

anthropology tend to analyze corruption at the level of individual in the context of societies (e.g., Smith 2008; Gupta 1995), whereas works in organizational anthropology often deal with corruption in the context of organizations (e.g., Jackall 1988). Lastly, human geography (particularly the critical strand) is different from anthropology despite its similar treatment of corruption as a social construction, as it focuses more on how different forms of corruption affects the lives of communities in relation to their respective socioeconomic statuses and access to resources (e.g., Walton 2013a).

Adapting and extending the work of Torsello and Venard (2016), the anthropological approach differs from and adds value to the rationalist literature in the economic and management/organization streams in the following ways. First, these studies ignore universal or formal definitions of corruption on the grounds that they fail to capture the complexities of the public and private categories prescribed in those definitions (Torsello and Venard 2016). Furthermore, they subscribe to the idea that the law is plural, it is not an objective entity, free from interpretations of the powerful (Pardo 2004). Consequently, these researchers are more interested in understanding social reality—how local communities define corruption—following the 'emic' approach in social research (Headland et al. 1990). This is consistent with sociological research, which argues that members of organizations or society have varying constructions of existing regulations which results in different ways of responding or complying with them (Gray and Silbey 2014).

The lack of interest in applying a strict definition of corruption has led anthropological scholars to argue that corruption is not inherently dysfunctional as most researchers believed. People may generally associate the word corruption with relatively similar notions like 'decay' or 'impurity' (Hindess 2012), but the practices labelled as such may be understood as something entirely different. In addition, they also problematize that certain definitions do not fit situated experience. Walton (2013b) for example, points out that Western interpretation of corruption obscures the experience of the poor and marginalized people of Papua New Guinea (PNG) insofar they see corruption as functional—it assists in securing their share of state resources.

Second, anthropological studies largely avoid moral evaluation and prefer multiple views of ethics and morality (Torsello and Venard 2016), following a social constructionist approach. They shy away from discussing corruption from an ethical or moral stance (except for few exceptions in which they dispute the objective treatment of the terms 'ethics' or 'moral'), because they are concerned with what they regard as a judgmental approach. The approach, taken by many rationalists, associates corruption with 'underdevelopment', 'poverty' and 'destructive behavior'. A more anthropological approach holds that this prevents an objective view of the socio-cultural complexities of corruption and that a judgmental evaluation of corruption limits the ability to understand certain practices and their local meanings, which need to be analyzed in context. For example, Gray (2013)'s analysis on the variety of techniques used to frame unethical actions as moral or justifiable indicates that there may be specific elements related to people's understanding of ethics or morality which give way to the perpetuation of certain corrupt practices that deserve further attention[2].

The anthropological view is more inclined towards understanding the complexity of ethics and (un)ethical behavior by paying attention to how they are grounded in people's situated experience (Carmalt 2011). This is consistent with the assertion by Shadnam (2014) that a homologous approach—one that treats morality and organization as socially co-constitutive—is important in studying ethics and morality.

Avoiding a moral evaluation resonates with the idea that scholars need to study the ethics and morality of corruption using a relational, grounded and situated approach because it allows researchers to capture the contingent nature of, and the complexities of, the social context involved in topics related

---

[2] What seems to be missing in Gray (2013)'s analysis is, however, the broader socioeconomic and cultural context within which those questionable actions take place, to which some anthropological works give more attention. In addition, the arguments put forward by Gray tends to overlook the kind of 'everyday corruption' (Nuijten and Anders 2017), which involves everyday citizens and the possibility of mindlessness as opposed to careful deliberation in explaining corrupt behavior.

to moral and ethics, including corruption (Clammer 2012). Instead of applying a fixed universal approach to ethics, the anthropological approach appreciates that ethics needs to be understood from the point of view of actors situated in a specific time and place. Adopting such a view has allowed sociologist (Ledeneva 2001) to unpack the complexity of *blat,* the use of personal connections in Russia which is often framed in a negative way because it bypasses formal procedures (Onoshchenko and Williams 2014). *Blat,* commonly labeled as corruption in Russia, is in fact just a different mode of exchange which does not carry any sense of moral decay (Ledeneva 2001).

Third, anthropological studies of corruption pay great attention to the processual aspect of corruption, particularly in how corruption is 'constituted' at a specific time and place. Instead of taking a static view, looking to establish whether or not corruption happens, they focus on the detailed processes involved whereby corruption comes about—a more processual view (Torsello and Venard 2016; Ashforth et al. 2008). Unlike rationalist scholars, who tend to reduce or collapse a series of unfolding events into what can be described as corruption into statistical summaries, they focus more on those very details. For example, in exploring corruption processually they elucidate the specific cultural processes and the complexities of people's experience of corruption and of the state in which choice of words plays a role in giving contour to those experiences (Gupta 1995).

Anthropological studies also attend to the interpretive and linguistic aspects of corruption as it views corruption as a "meaningful, culturally constructed, discursively mediated, symbolically saturated, and ritually regulated" (Brubaker and Laitin 1998, p. 441) social phenomenon. This means that researchers adopting an anthropological approach will not look at corruption as an objectively identifiable phenomenon. Nor will they view corruption as merely a set of social practices. Instead, they will also attend to the textual aspects of those practices, following the language and meanings that social actors attribute to it (Torsello 2010). The anthropological approach also resembles many of the features of the homologous view of corruption (Shadnam 2014) as they view corruption as a phenomenon inseparable from the social dimensions of human behavior, and which continues to be defined and re-defined through everyday communicative practices.

Corruption could not and should not be separated from patterns of thought and action which sustain it by way of social regulations and sanctions. An anthropological approach is interested in investigating not the 'effects' of corruption but the 'constitution' of corruption in a specific time and place (Shadnam 2014). In exploring corruption processually, Orjuela (2014) describes how corruption enables people to maintain a sense of ethnic stability and perseverance. Using examples from Nigeria, Kenya and Sri Lanka, she underscores the complexities of corruption beyond cost-benefit calculations, illuminating individual motivations and struggles to fulfil ethnic identity expectations. In paying attention to the language and meanings that people use to describe corruption, she captures the complexity of corruption as a political project through which people strive to be seen as loyal to their community.

Lastly, the anthropological approach draws attention to the role of meanings and identities embedded in social practices, many of which are labelled as corruption (Misangyi et al. 2008). Because of the different assumptions explained previously, anthropological studies of corruption engage more deeply with the understanding of the subjective experience of individuals, their ways of being and doing things, which leads to an exploration of meanings of various practices and role identities of people involved in those practices (Torsello and Venard 2016). Such a view has enabled scholars to demonstrate how role identities shape and are shaped by people's experience of corruption. More importantly, in examining identities, the approach may help explain the normalization of controversial practices which allow "good people do bad things" (Kaptein 2013). Breit (2011), for instance, shows that in the case of an alcohol monopoly scandal in Norway, a country that is relatively free from corruption according to Transparency International, corruption not only allows people to attack controversial business practices, but also to express a critique towards the state's dominant role in the alcohol market and a way of articulating ideas about Norwegian national moral identity. These studies show how the alternative understandings of corruption are achieved by placing the cultural

context in the foreground of corruption research. These alternative understandings are important, not only because they help us to rethink about existing anti-corruption efforts, but also because they help us to think about delicate and overarching social and political issues from which corruption cannot be separated.

## 5. Conclusions

This paper has so far contributed by connecting various important research findings in relation to corruption and recognizing their different approaches. It has discussed the rationalist approaches to studying corruption that tend to treat corruption as universally and inherently 'destructive'. It then identified three central debates within the management and organization literature that concern:

(1) Whether corruption is a result of mindless or mindful processing; (2) the role of individual's perception or subjectivity and (3) the role of social contexts or environment in influencing their perception. In addressing the debates, the paper has drawn from anthropological studies (Torsello and Venard 2016; Haller and Shore 2005) which emphasize a contextualized understanding of corruption, which eschew formal definitions and moral evaluations, and which focus attention on process, meaning and identity.

Consequently, upon applying insights from anthropological and related studies to address the debates within organization and management works, this paper proposes a cross-disciplinary approach to studying corruption. Such study will focus on the interpretation of corruption by organization members situated in context as well as the various identities and processes through which corruption comes to be experienced by individuals. It will ask, for example, questions such as "*What does corruption mean to people in this particular organization or community?*" and "*How do people construct or interpret different issues in relation to various practices associated with corruption?*". In exploring identities, it will ask questions such as "*What role identities are being invoked when people discuss their experiences of corruption?*". With regards to the view of multiple morality, the study will explore the issue by asking questions such as: "*What does it mean to be moral or ethical to people in this specific organization or community?*" and "*How do notions of ethics and morality feature in the constructions of corruption?*". To allow such an endeavour, researchers will need to adopt a research approach that researchers in anthropology and related field have already been familiar with. For example, they may apply qualitative methodology and methods such as ethnography, interviews or media text analysis, and adopt an interpretive perspective—which takes human interpretation as their starting point for understanding the social world (Burrell and Morgan 1985; Moran 2002). Asking the above basic and open-ended questions on meanings and interpretations would allow researchers to return to the three debates outlined previously.

In addition to addressing the debates, a cross-disciplinary approach will potentially contribute to the literature in at least three ways. First, by asking questions around people's intersubjective meanings and experiences of corruption, researchers are able to explore the perspective of the 'insiders', one that is often silenced in empirical studies on corruption. Mainstream research has largely treated corruption as having a singular, pre-determined and fixed meaning across contexts (Martin and Parmar 2012; Sonenshein 2007). It tends to treat social actors as passive entities and to see their own task as being to uncover this pre-existing meaning and behave accordingly (Martin and Parmar 2012). However, a cross-disciplinary approach treats corruption differently—as a socially constructed phenomenon. In this way, it has the potential to enrich or even challenge existing research by showing the multifarious nature of corruption.

Second, a cross-disciplinary approach that eschews from moral evaluation has the potential of bringing in the views of certain members of society, such as public servants, politicians, the poor and the uneducated, who are often barraged for their 'corruptness'. Clearly, people's understanding of corruption is influenced by their intersubjective experience of the phenomenon (Znoj 2007). A cross-disciplinary approach will address the limited attention paid to the lived

experience of corruption, especially in management and organization studies (Sonenshein 2007; Torsello and Venard 2016).

Third, by introducing a cross-disciplinary approach that appreciates individual interpretation of social issues, researchers will expand the conceptual tool box of corruption research in management and organization studies by addressing what some scholars have identified as a limitation of corruption theories in management and organization studies: the lack of "research from a 'local' point of view" (Torsello and Venard 2016, p. 50).

Notwithstanding all of the potential theoretical contributions above, applying anthropological insights into organization research is not without its limitations. Researchers must be aware that the problem of access is one of the first barriers. Given the general moralizing tendency, it is important, but understandably difficult, for researchers to identify potential research participants who would be comfortable sharing their experiences of practices which many often label as corruption. In addition, this type of research brings the risk of researchers being exposed to illegal practices and this posits potentially difficult questions about legal responsibility. This is especially true in the light of a researcher's obligation to protect their participants from harm. Finally, there is also limitation in terms of reporting back the findings to the research community in which books are more preferred compared to journal articles due to the massive amount of data that needs to be presented and analyzed (Torsello and Venard 2016).

Finally, a cross-disciplinary research has important potential contributions for anti-corruption practices. Learning that corruption might be treated as "unwritten rules" (p. 2), a different mode of interpersonal exchange which does not carry any sense of moral decay (Ledeneva 2001), it became evident that there is a wide gap to bridge between those standing on behalf of anti-corruption campaigns and good governance and those whose practice are being scrutinized. Moreover, in light of limited achievements of the global anti-corruption movement, an understanding of the meanings of practices many labelled as corruption in its specific context may help to reveal the limitations of existing approaches. Through exploring meanings in context, anti-corruption and good governance campaigners may begin to evaluate whether their existing strategies speak to the communities they seek to shape or influence.

**Acknowledgments:** The author acknowledges the support of the Endeavour Awards and the University of Melbourne, which aided the research for this study and would like to thank the editor and the two anonymous reviewers for their helpful comments and suggestions.

**Conflicts of Interest:** The author declares no conflict of interest.

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
