# Peer review of "Contextualizing Corruption: A Cross-Disciplinary Approach to Studying Corruption in Organizations"

_admsci, doi:10.3390/admsci8020012_

Round 1

Reviewer 1 Report

Overall, an interesting paper with an important message:

That the study of corruption/ethics would benefit from a social construction approach that takes into consideration the social meanings people attach to their behaviors and how ethics and corruption is often formed through social interactions. There is value to researching insider perspectives on corruption.

In order to better understand the above approach, organization and management scholars should consider what the author(s) refer to as ‘the anthropological approach’ (which includes, according to the author, the fields of geography, discourse and human ethics).

Now, given that the author(s) point out the importance of understanding social context, social meanings, and social interactions one is left to wonder why ‘sociology’ has been ignored in this article?

The focus that the author argues for is basically what many sociologists already do in their research – in other words, it seems that the author is advocating for greater sociological engagement in the field of organization and management studies.

Sociology “IS” the study of how individuals attach social meanings to their behaviours. A social constructionist perspective is also used and has been most developed in the field of sociology (as simply one of several different theoretical paradigms that sociologists draw upon). To see an example in the field of organization and management studies, see the work of Nick Turner (in particular, his special issue on “Socially Constructing Safety” in the management journal, ‘Human Relations). Also, if the author(s) wish to include anthropology, then, it would be wise to specify: ‘cultural’ anthropology might be more appropriate than simply stating an anthropology.

Overall, a great deal of what is discussed in this article (countering the rationalist perspective) has been the focus of past sociological research and debates (from the work of Diane Vaughan to Charles Perrow – two very influential sociologists whose work has implications for organization and management studies). The sociology of work and organizations has a long history that is completely neglected in this article.

While (cultural) anthropology has much to offer management research so too does the field of organizational studies in sociology. It is confusing as to why anthropology is the focus here (or geography for that matter).

There is also value to looking to the field of criminology. For instance, see Gray’s ‘Insider Accounts of Institutional Corruption: Examining the Social Organization of Unethical Behavior’ (British Journal of Criminology) as well as the work by Gray and Susan Silbey (2014, American Journal of Sociology) on the importance of understanding how people interpret and experience rules and laws. Both the above articles (in criminology and sociology) take on much of what the author suggests is needed in organization and management fields.

The frustration that the author may feel (that management studies focuses primarily on the individual opposed to the social context in which an individual is located) is not new. It is a debate that continually plays out in sociology.

However, despite the limitations above, this reviewer still believes that there is value in what the author(s) is trying to do, namely, advocating that organization and management studies move beyond a strict rationalist approach.  However, as noted, this is not a new idea. And, as noted, there is research that does attempt to do this in the field of ethics and corruption.

I recommend that the author(s), in their quest to push for a cross-disciplinary approach, expand the literature and disciplines that are included in their analysis.

Best wishes as you continue to develop your interesting manuscript!

Author Response

First, I would like to thank Reviewer 1 for taking their time to review this paper and provide helpful suggestions.

Second, I acknowledge their concerns regarding the inclusion of sociology literature in the paper although I understand that there are at least four works that have been cited that are associated with the field of sociology as follows:

Brubaker, Rogers, and David D Laitin. 1998. "Ethnic and nationalist violence."  Annual Review of Sociology:423-452.

Burrell, Gibson., and Gareth Morgan. 1985. Sociological paradigms and organisational analysis : elements of the sociology of corporate life. Aldershot, Hants.: Gower.

Clammer, John. 2012. "Corruption, development, chaos and social disorganisation: Sociological reflections on corruption and its social basis." In Corruption: Expanding the focus. ANU Press.

Ledeneva, Alena. 2001. "Unwritten rules."  London: Centre for European Reform.

However, I still found their suggested papers to be very useful and I have included them in the revised paper where appropriate.

Third, I fully agree with them that sociology has a great deal of influence in the study of management/organization which may explain the interpretive turn in the field. My reason for not examining the sociology literature more extensively from the very beginning and not using sociology as a particular benchmarking category is that because the anthropological literature provides a more direct comparison and shares more similarities in terms of vocabulary (e.g., culture, identity) with management works.

The sociological works that the reviewer suggested, for example, did not discuss corruption in a manner that is in parallel to how management/organization scholars have analyzed it. Meanwhile, discussions about the plurality of norms or laws in anthropology are more readily transferable to a management audience interested in corruption research. Nevertheless, as stated, I found the works suggested (including the criminology paper) to be helpful and therefore I have included them and few others to strengthen the overall paper.

Fourth, I chose to use “anthropology” and not “cultural anthropology” for simplicity reason because this paper is targeted at the broader audience in the field of management given the use of anthropology in management theorizing is still quite limited, particularly when compared to the more mainstream approach to studying corruption. In my understanding, to specify it as “cultural anthropology” may risk complicating the argument unnecessarily. Regarding geography, I have revised by using "human geography" to refer to some works on corruption which emphasize the human aspects of corruption such as the works of Walton (2013) cited in the paper.

Finally, I appreciate the reviewer’s encouragement on what the paper is trying to achieve, and I understand that there is space for the paper to advocate for a cross-disciplinary approach by showing the tensions within the management/organization field itself. The paper will hopefully give a stronger push given that there is still very limited work that examined what corruption means from the perspective of the ‘insiders’.

Reviewer 2 Report

The present paper provides an overview of corruption literature and links different approaches of corruption with an objective of providing a guide to organization and management research to apply insights from other disciplines with an emphasis on anthropological approach.

The paper is well written and provides a good overview of existing literature from the points of view of different disciplines. It does, however, sometimes cites very old papers and does not mention new studies that have validated/invalidated the older findings with better data and/or methodology and/or added to this literature by adding a different perspective. I will talk about of some of these studies based on my own expertise. For instance, the paper talks about women being less tolerant in some countries than other suggesting that the relationship between gender and corruption may be culture specific (Alatas et al., 2009). While this is true, several recent papers have studied these issues in a greater detail and have important findings which should be mentioned in the paper. For instance, Jha and Sarangi (2015) using IV analysis to address endogeiniety concerns find that its women’s presence only in politics that reduces corruption suggesting that women may not inherently be less corrupt. Similarly, the paper also discusses the impact of culture (Universalist vs relativist morality and cultural dimensions such as power distance index and masculinity) on corruption but does not mention most recent studies that may be helpful for readers and researchers. For instance, Jha and Panda (2017) show that more individualist countries are less corrupt than collectivist countries. Some of the studies that have been cited relating this literature are more than a decade old.

Of course, there might be some other areas with these issues but I may not be aware of the recent developments in those areas.

I found a minor typo: on page 8 in first paragraph: “A market that is ….. in the hands of a number wealthy families” – “of” is missing in that sentence between number and wealthy.

Bottom line: I see this paper more as an overview of the literature paper than addressing some research questions. The study therefore should update their references to be more useful to readers and academic community.

References

1.       Esarey, J., & Schwindt-Bayer, L. (2017). Estimating Causal Relationships Between Women’s Representation in Government and Corruption.

2.       Jha, Chandan Kumar and Sudipta Sarangi. (2015). “Women and Corruption: What Positions Must They Hold to Make a Difference?”. Available at SSRN:  https://papers.ssrn.com/sol3/papers.cfm?abstract_id=2434912  

3.     Jha, C., & Panda, B. (2017). Individualism and Corruption: A CrossCountry Analysis. Economic Papers: A journal of applied economics and policy1(36), 60-74.

Author Response

I would like to thank Reviewer 2 for taking their time to review this paper and provide helpful suggestions.

Following up on their comments, I have added several new (more updated) references including the ones mentioned by the reviewer to enrich the paper. 

I have also fixed the typo in the revised file.